

# Identification of the hub genes in gastric cancer through weighted gene co-expression network analysis

Chunyang Li[1,2], Haopeng Yu[1,2], Yajing Sun[1,2], Xiaoxi Zeng[1,2] and Wei Zhang[1,2]

[1] West China Biomedical Big Data Center, West China Hospital, Sichuan University, Cheng, China
[2] Medical Big Data Center, Sichuan University, Chengdu, China

## ABSTRACT

**Background**. Gastric cancer is one of the most lethal tumors and is characterized by poor prognosis and lack of effective diagnostic or therapeutic biomarkers. The aim of this study was to find hub genes serving as biomarkers in gastric cancer diagnosis and therapy.

**Methods**. GSE66229 from Gene Expression Omnibus (GEO) was used as training set. Genes bearing the top 25% standard deviations among all the samples in training set were performed to systematic weighted gene co-expression network analysis (WGCNA) to find candidate genes. Then, hub genes were further screened by using the "least absolute shrinkage and selection operator" (LASSO) logistic regression. Finally, hub genes were validated in the GSE54129 dataset from GEO by supervised learning method artificial neural network (ANN) algorithm.

**Results**. Twelve modules with strong preservation were identified by using WGCNA methods in training set. Of which, five modules significantly related to gastric cancer were selected as clinically significant modules, and 713 candidate genes were identified from these five modules. Then, *ADIPOQ*, *ARHGAP39*, *ATAD3A*, *C1orf95*, *CWH43*, *GRIK3*, *INHBA*, *RDH12*, *SCNN1G*, *SIGLEC11* and *LYVE1* were screened as the hub genes. These hub genes successfully differentiated the tumor samples from the healthy tissues in an independent testing set through artificial neural network algorithm with the area under the receiver operating characteristic curve at 0.946.

**Conclusions**. These hub genes bearing diagnostic and therapeutic values, and our results may provide a novel prospect for the diagnosis and treatment of gastric cancer in the future.

## BACKGROUND

Gastric carcinoma remains the fifth most frequently diagnosed cancer and the third leading cause of cancer-related deaths, with an estimated 1,033,701 new cases and 782,685 deaths worldwide in 2018 (*Bray et al., 2018*; *Pormohammad et al., 2018*). Gastric cancer is also one of the most common malignancies and the third leading cause of death in China, where 427,100 cases with 301,200 deaths were observed in 2013 (*Chen et al., 2017b*). Despite the

Corresponding author
Xiaoxi Zeng, zengxiaoxi@wchscu.cn

several existing treatments including chemo-, radio-, or targeted therapy, the overall 5-year survival rate of stomach cancer patients is still <20% (*Raimondi et al., 2018*).

There are two types of gastric cancer, diffuse and intestinal types, which differ in their histological manifestations, epidemiological features and etiologic pathogenesis (*Huang et al., 2019*). Histopathology is the gold standard approach for diagnosing gastric cancer; however, this approach is not suitable for everyone due to the invasive nature of the biopsy (*Yoon & Kim, 2015*). Although there are several commonly used serum biomarkers such as alpha-fetoprotein (AFP), carcinoembryonic antigen (CEA), cancer antigen 125 (CA125), and cancer antigen 19–9 (CA19-9) (*He et al., 2013*) for gastric cancer diagnosis, none of them are sensitive or gastric-cancer-specific (*Smyth et al., 2016*). Moreover, effective and specific targeted therapies for gastric cancer remain to be identified. Presently, the major treatment strategies for gastric cancer are anti-human epidermal growth factor receptor 2 (HER2) and anti-vascular therapies (*Raimondi et al., 2018*). However, resistance to the targeted agents is common in some gastric tumor types. Therefore, novel practical approaches are needed for specific diagnosis and effective treatment of gastric cancer. Accordingly, identification of the key genes and biomarkers that are involved in the pathogenesis of gastric cancer is of paramount significance.

With recent advancements in bioinformatics methods, comprehensive identification of potential biomarkers through large-scale screening of expression profiles has been proposed (*Li et al., 2018a*; *Takeno et al., 2008*; *Wang et al., 2014*; *Zeng et al., 2019*). A weighted gene co-expression network analysis (WGCNA) approach provides a systematic analysis to investigate the functional clustering of expression profiles, based on the theory that genes with similar expression profiles may have closely functional linkages and/or pathways (*Carlson et al., 2006*; *Carter et al., 2004*; *Zhou et al., 2018*). This approach groups highly co-expressed genes into the same module. Modules bearing high correlation with certain clinical traits are identified as clinically significant modules (*Zhou et al., 2018*).

By using this systematic bioinformatic method, followed by the "least absolute shrinkage and selection operator" (LASSO) logistic regression, a suitable method for high-dimensional gene data analysis (*Friedman, Hastie & Tibshirani, 2010*; *Zeng et al., 2019*), candidate variables were selected from clinically significant modules. Finally, supervised artificial neural network (ANN) method was performed to test the reliability of the results in an independent dataset. ANN approach has been widely used in the prediction of cancer diagnosis, staging and recurrence since the mid-1990s (*Hu et al., 2013*), which is an useful method to incorporate and analyze large amounts of omics and health-care data (*Ngiam & Khor, 2019*).

Consequently, we attempt to construct a co-expression network by using systematic WGCNA method followed by LASSO regression to identify hub genes, which could effectively discriminate cancer samples from normal tissue. These findings may provide potential diagnostic and therapeutic targets in future research and clinical intervention of gastric cancer.
## MATERIAL AND METHODS

### Data collection and preprocessing

The workflow of this study is shown in Fig. 1. Raw expression datasets were downloaded from the Gene Expression Omnibus (GEO) database (http://www.ncbi.nlm.nih.gov/geo/) by using the keywords "stomach/gastric cancer/tumor/carcinoma", "normal", "GPL570", and "*Homo sapiens*". Our inclusion criteria for the training set were that: (1) datasets based on the Affymetrix Human Genome U133 Plus 2.0 Array Platform (Affymetrix, Santa Clara, CA, USA) and (2) datasets derived from human case-control studies, with gastric tumor patients as the case group, regardless of the histopathological types and stages, and non-tumor individuals as the control group.

Therefore, only two datasets (GSE66229 (*Cristescu et al., 2015*; *Oh et al., 2018*) and GSE54129) as of October 10, 2020 met the screening criteria. GSE66229 contained 300 tumor and 100 normal samples, and was used as the training set to screen for the hub genes. GSE54129 (contained 111 tumor and 21 normal samples) served as an independent testing set to validate the hub genes.

All the analyses in this study were conducted using R software (version 3.5.1). FitPLM weight, Relative Log Expression (RLE), Normalized Unscaled Standard Errors (NUSE), and RNA degradation images were analyzed to evaluate the quality of each dataset. Then, the "rma" function with the default parameters of the "affy" package was used to perform background correction and normalization (*Gautier et al., 2004*). Missing values in each dataset were imputed by using the function "impute.knn" with the default parameters of the "impute" package (*Hastie & Narasimhan, 2001*). Platform annotations were downloaded from the GEO database, and finally, the gene symbol expression matrices were acquired from each dataset for further analyses.

### Weighted gene co-expression network construction

Weighted gene co-expression network in the training set was constructed using the "WGCNA" package (*Langfelder & Horvath, 2008*; *Zhang & Horvath, 2005*). The genes with the top 25% SD among all the 400 samples in the expression matrix of the training set were selected as the input genes (5,115 genes in total).

In brief, first, the appropriate soft-thresholding power ($\beta$) was selected by using the "pickSoftThreshold" function with the default parameters (herein, $\beta = 4$). Subsequently, the Pearson's correlation matrix was calculated to evaluate the similarity among all the pair-wise genes by using the "cor" function with the default parameters. Then, the adjacency was calculated based on $\beta$ and the Pearson's correlation matrix by using the "TOMsimilarity" function with the default parameters, and the corresponding dissimilarity (dissTOM) was also calculated. Finally, average linkage hierarchical clustering was conducted according to the dissTOM value with a minimum size of 30 for each gene dendrogram.

Module eigengenes (MEs), considered the first principal component (PC) of gene expression patterns of a corresponding module, were obtained for each module. To further strengthen the reliability of the modules, a cut line was set at 0.25 so that modules bearing <0.25 would be merged (*Chen et al., 2017a*).
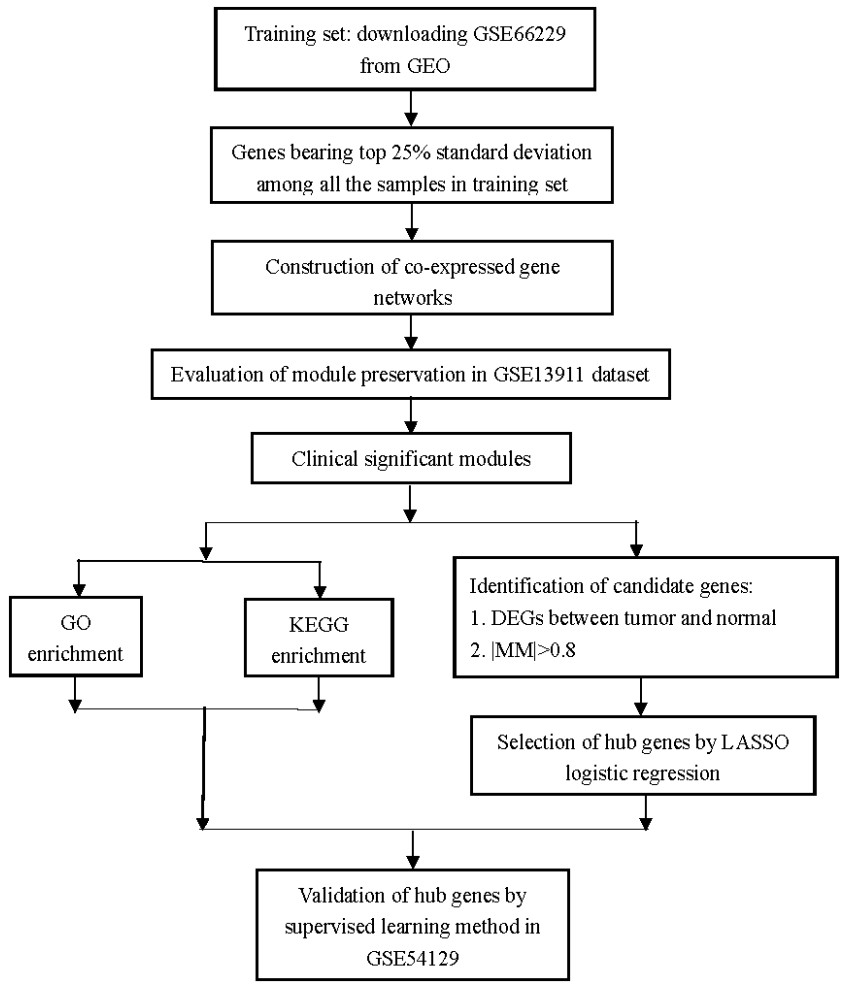

**Figure 1** **Flow diagram of the study.** Data collection, analysis, and hub gene selection and validation.

## Module preservation analysis

To evaluate the stability of the modules in the training set, GSE13911 was used to validate the module preservation of the training set (*Chen et al., 2018*; *Neidlin, Dimitrakopoulou & Alexopoulos, 2019*; *Obeidat et al., 2017*). Preservation analysis for GSE13911 was performed using the "modulePreservation" function by setting referenceNetworks = 2, nPermutations = 200, randomSeed = 1, and verbose = 3, maxModuleSize = 3000 and maxGoldModuleSize = 3000 in "WGCNA" package. The $Z$ summary scores of each module were calculated to indicate the module preservations. $Z$ summary scores $<2$, [2–10], and $>10$ indicated that the modules had no, moderate, and strong preservation, respectively (*Liu et al., 2019b*; *Lou et al., 2017*). The grey module contained the genes that did not belong to any of the other modules, and a gold module was generated for statistical purposes. Therefore, these two modules were not shown in the preservation analysis results (*Langfelder et al., 2011b*).

### Identification of clinically significant modules

Herein, the most interesting clinical trait was the tissue type, which was designated as tumor or normal samples. We calculated $\log_{10}$ transformation of the *P*-value in the logistic regression between the MEs and clinical trait. Modules with $\log_{10}$ transformation of the *P*-value greater than 10 were considered to be closely correlated with tissue types.

### Functional enrichment and pathway analyses of significant modules

To determine whether the clinically significant modules were closely correlated with gastric cancer, GO functional annotation and KEGG pathway analyses were performed using the Database for Annotation, Visualization and Integrated Discovery (DAVID) (version 6.8) (https://david.ncifcrf.gov/home.jsp) (*Ashburner et al., 2000*; *Dennis et al., 2003*). The visualization of the functional enrichment and pathway analyses was performed by the "GOplot" (*Robin et al., 2011*) and "ggplot2" (*Wickham, 2016*) packages of R, respectively.

### Candidate gene selection

Candidate genes among clinically significant modules were selected according to the following criteria: (1) differentially expressed genes (DEGs) between gastric cancer samples and normal samples with $|\log_2 FC\ (Fold\ Change)|>1$ and adjusted *P*-value <0.05 based on the "limma" package (*Ritchie et al., 2015*), (2) high module membership (defined as the correlation between the expression of each genes and MEs) |MM(Module Membership)|>0.8.

### Selection of hub genes by LASSO logistic regression analysis

Candidate genes were subjected to LASSO regression, which was performed using the "glmnet" package by setting alpha = 1, and ten-fold cross-validation for tuning parameter selection. Lambda was defined as the minimum partial likelihood deviance (*Friedman, 2010*).

### Validation of hub genes

The machine learning method of ANN (by using the "neuralnet" function with hidden = 2) (*Fritsch et al., 2019*) was performed to determine whether the hub genes could correctly distinguish the gastric cancer samples from the normal samples in a testing set. Moreover, in order to demonstrate whether these hub genes could specifically distinguish between gastric tumor and normal samples, we also evaluated the predictive effects of 11-gene model in pancreatic cancer (GSE15471) (*Badea et al., 2008*) and colorectal cancer (GSE37364 excluding adenoma samples) (*Galamb et al., 2012*) by using ANN algorithm.

Areas under the receiver operating characteristic (ROC) curve was calculated to show the predictive effect of supervised machine learning model, and then the ROC curves were plotted using the "pROC" package (*Robin et al., 2011*). An area under the curve (AUC) value between 0.8 and 0.9 is considered an excellent classification, while greater than 0.9 is considered as outstanding discrimination (*Lemeshow, 2000*).
## RESULTS

### Construction of co-expression networks

After the quality check of the input data, no sample was removed (Fig. S1); herein, two clinical traits (tissue type and stage) are presented. According to different tissue types (tumor or normal), the 400 samples could be mainly divided into two clusters.

As shown in Fig. 2, the soft thresholding was set at 4, while the scale-free topology fit index reached 0.89, indicating approximate scale-free topology. Co-expressed gene modules were identified with the dynamic tree cut method (Fig. 3A) (*Chen et al., 2017a*). In total, 12 modules were found, and each color represented one module (Fig. 3B). The biggest module was the turquoise module, which contained 2,120 genes, followed by the blue module, bearing 1,503 genes. The grey module comprised 2 genes, which did not have a similar expression pattern and did not belong to any other module.

### Module preservation analysis

5,115 genes in GSE13911 clustered into 11 colored modules (grey module only contained 2 genes, and did not show in Fig. 4), as determined in the training set. All gene modules were found to bear strong conservation, as the $Z$ summary scores were all >10 (Fig. 4B).

### Selection of clinically significant modules

After the assessment of the relationship by using regression analysis between the MEs and clinical traits, the $\log_{10}$ transformation of the $P$-value was shown in Fig. 5. Accoring to the screening criteria, there were 5 modules closely related to tissue types, which were black, turquoise, greenyellow, salmon and blue modules. These moduels were selected for further analysis.

### Functional enrichment of clinically significant modules

Gene Ontology (GO) enrichment results showed that 1503 genes in the blue module, 137 genes in the black module, 2,120 genes in the turquoise module, 61 in the salmon module and 194 genes in the greenyellow module mainly participated in 141, 15, 221, 54 and 146 different significant biological processes, respectively (Table S1 to Table S5). The top three most significantly enriched biological processes were cell division ($P = 2.29e-29$), G1/S transition of the mitotic cell cycle ($P = 6.32e-21$), mitotic nuclear division ($P = 3.49e-19$) in the blue module (Fig. 6A), and potassium ion import ($P = 3.39e-05$), digestion ($P = 8.25e-04$), multicellular organismal water homeostasis ($P = 0.0012$) in the black module (Fig. 6B), and extracellular matrix organization ($P = 6.30e-06$), positive regulation of cell migration ($P = 7.57e-06$), axon guidance in the turquoise module ($P = 9.18e-06$) (Fig. 6C), and inflammatory response ($P = 2.80e-21$), immune response ($P = 5.70e-17$), neutrophil chemotaxis ($P = 5.15e-15$) in the greenyellow module (Fig. 6D), and defense response to virus ($P = 3.33e-25$), type I interferon signaling pathway ($P = 8.52e-23$), response to virus ($P = 9.02e-16$) in salmon module (Fig. 6E).

KEGG (Kyoto Encyclopedia of Genes and Genomes) analysis showed that genes in the blue, black, turquoise, salmon and greenyellow modules were mainly significantly enriched in 15, 7, 35,14 and 23 pathways, respectively (from Table S6 to Table S10).
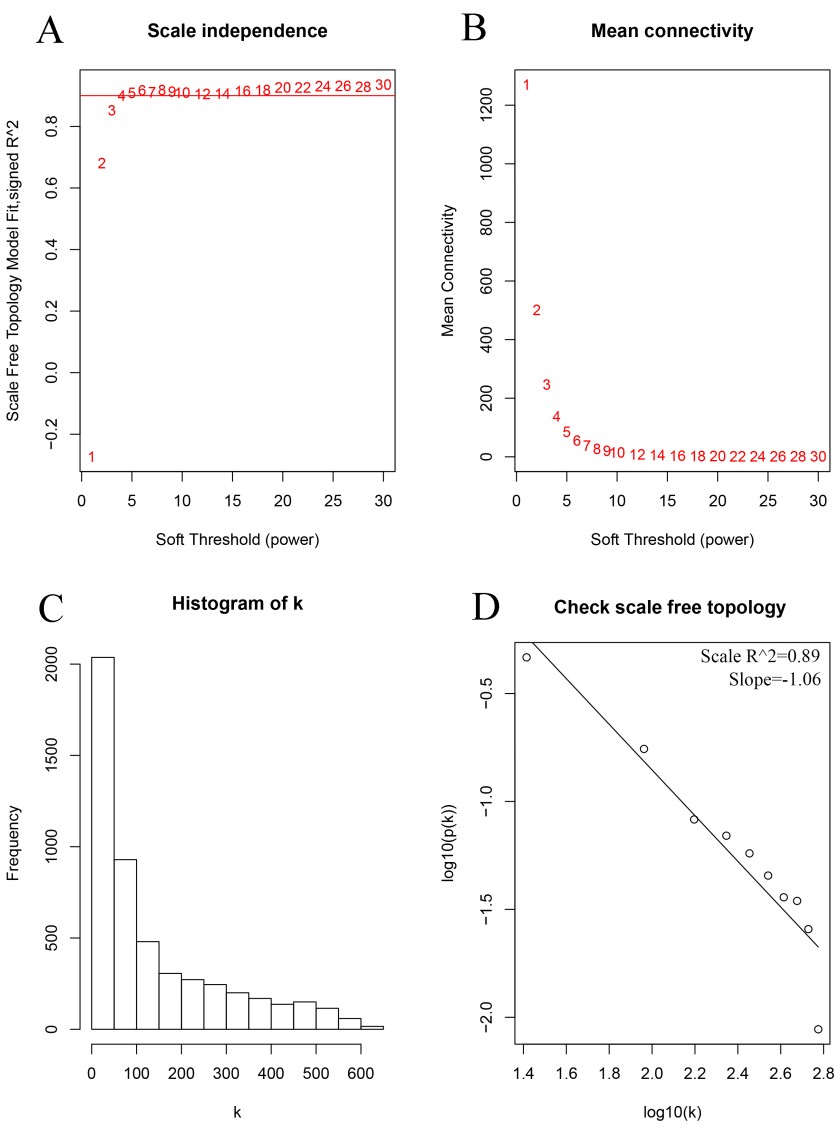

**Figure 2  Determination of the soft-thresholding power in the weighted gene co-expression network analysis in the training set.** (A) Screening soft-thresholding powers. (B) Analysis of the mean connectivity for various soft-thresholding powers. (C) Histogram of the connectivity distribution with the soft-thresholding powers set at 4. (D) Checking the scale-free topology with the soft-thresholding powers set at 4.

The top three most significantly enriched pathways were cell cycle ($P = 1.21e{-}15$), DNA replication ($P = 6.21e{-}15$), the p53 signaling pathway ($P = 2.03e{-}06$) in blue module (Fig. 7A), and gastric acid secretion ($P = 4.46e{-}05$), protein digestion and absorption ($P = 1.28e{-}04$), drug metabolism-cytochrome P450 ($P = 0.0032$) in black moule (Fig. 7B), and focal adhesion ($P = 1.49e{-}07$), arrhythmogenic right ventricular cardiomyopathy ($P = 2.27e{-}05$), complement and coagulation cascades ($P = 3.58e{-}05$) in turquoise module (Fig. 7C), and cytokine-cytokine receptor interaction ($P = 2.35e{-}08$), amoebiasis ($P = 3.30e{-}05$), and rheumatoid arthritis ($P = 5.65e{-}05$) in greenyellow

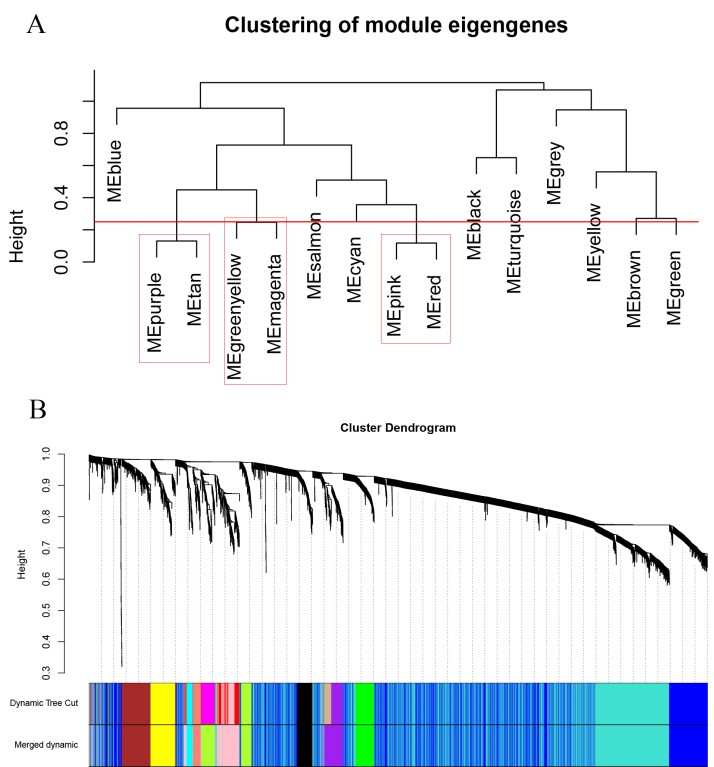

**Figure 3 Clustering dendrograms of the 5,115 genes in the training set.** (A) Clustering of the module eigengenes to identify the merged modules. Upon setting the threshold at 0.25, 15 modules were merged into 12 modules. (B) Co-expression module of the training set.

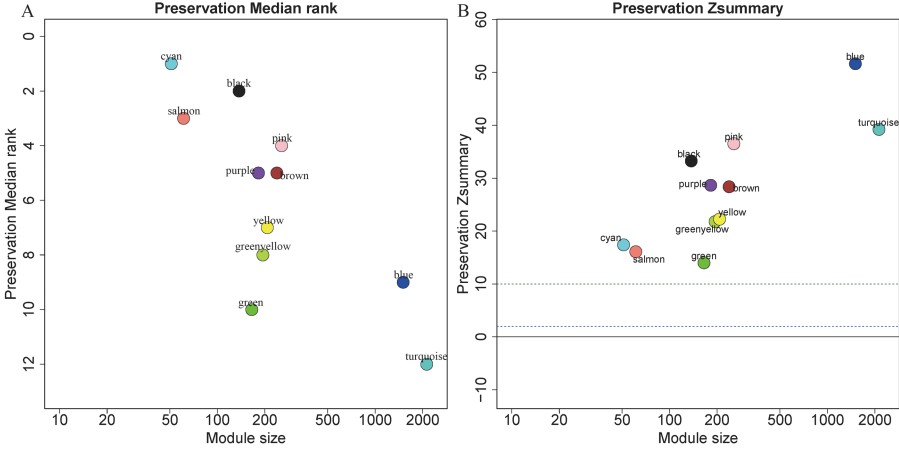

**Figure 4 Evaluation of module preservation.** The *x*- and *y*-axes present module size and preservation median rank (A) as well as preservation Z summary (B), respectively.

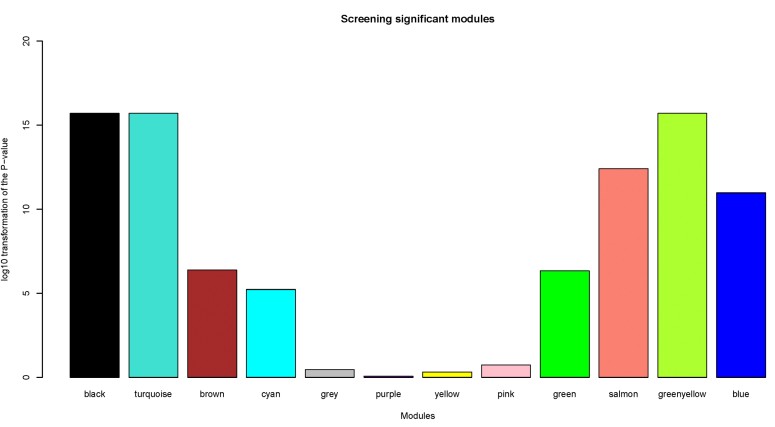

**Figure 5  Identification of clinically significant modules.** $\log_{10}$ transformation of the $P$-value in the logistic regression between the MEs and clinical trait. The height of bars represents the $\log_{10}$ transformation of the $P$-value, and modules with $\log_{10}$ transformation of the $P$-value greater than 10 were considered to be closely correlated with tissue types.

module (Fig. 7D). While in the salmon module, influenza A ($P = 1.09\text{e}{-}09$), measles ($P = 9.44\text{e}{-}07$), and herpes simplex infection ($P = 7.99\text{e}{-}06$) (Fig. 7E) were the most significantly enriched pathways.

## Identification of hub genes

Using the screening criteria of |MM |>0.8, 926 genes were identified from 5 significant modules, of which 713 genes were differentially expressed between the normal and tumor samples with |logFC|>1 and adjusted $P$-value <0.05 (all the 713 candidate genes were listed in Table S11). Finally, 11 genes [Adiponectin (*ADIPOQ*); Rho GTPase activating protein 39 (*ARHGAP39*); ATPase family AAA-domain containing protein 3A (*ATAD3A*); *C1orf95* (also known as *STUM* gene); Cell wall biogenesis 43 C-terminal homolog (*CWH43*); Glutamate receptor, ionotropic kainate 3 (*GRIK3*); Inhibin subunit beta A (*INHBA*); sodium channel epithelial 1 subunit gamma (*SCNN1G*); Sialic acid-binding immunoglobulin-like lectin-11 (*SIGLEC11*); Retinol dehydrogenase 12 (*RDH12*) and lymphatic vessel endothelial hyaluronan receptor 1 (*LYVE1*)] were identified as the hub genes by using LASSO logistic regression (Table 1). The heatmap of these 11 hub genes were shown in Fig. 8, indicating that these 11 hub genes differentially expressed between tumor and normal samples.

## Validation of the hub genes

GSE54129 was utilized as the testing set to validate the 11-gene model. The AUC value of this classifier upon using artificial neural network was 0.946 indicating the excellent classification effects of the model (Fig. 9A). Furthermore, both the AUC values of this 11-gene model were around 0.5 in colorectal cancer (Fig. 9B) and pancreatic cancer (Fig. 9C), indicating specifically predictive effect in gastric cancer.

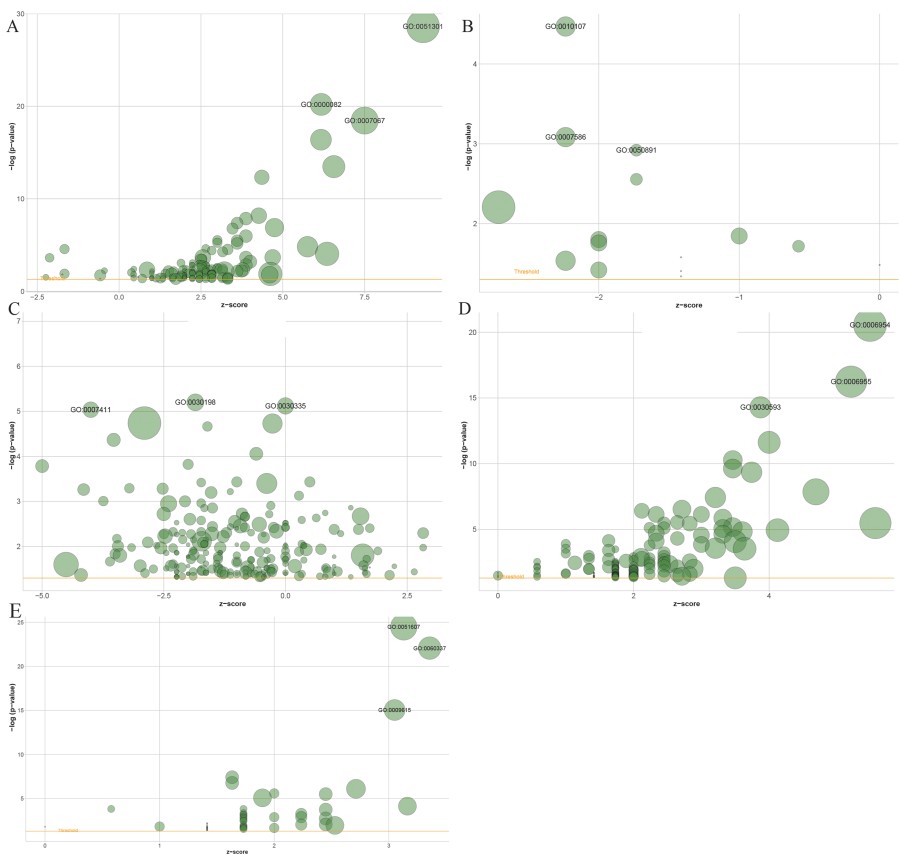

**Figure 6** **The bubble plot of gene ontology terms in the (A) blue, (B) black, (C) turquoise, (D) greenyellow and (E) salmon modules.** The $z$-score is assigned to the $x$-axis and the negative logarithm of the $p$-value to the $y$-axis, as in the barplot (the higher the more significant). The area of the displayed circles is proportional to the number of genes assigned to the terms. Herein, only the meaningful enriched GO terms were presented, and the most three sigificant GO terms' lables were displayed.

## DISCUSSION

In this study, five modules were identified as clinically significant and preserved modules by using WGCNA. The GO and KEGG analyses revealed that the genes in these four modules were significantly enriched in the biological processes of the cell cycle, cell division, and stomach-related functions. All these biological functions are closely related to gastric cancer (*Cao et al., 2018*; *Waldum, Sagatun & Mjones, 2017*). Eventually, 11 hub genes including *ADIPOQ*, *ARHGAP39*, *ATAD3A*, *C1orf95*, *CWH43*, *GRIK3*, *INHBA*, *RDH12*, *SCNN1G*, *SIGLEC11* and *LYVE1* were screened by using WGCNA method followed by LASSO regression. Then, artificial neural network algorithms were performed, and demonstrated that this 11-gene model could effectively discriminate between gastric cancer and normal tissues.

In preservation analysis, Z summary is used to assess the significance of observed statistics and is defined as the mean of Z scores computed for density and connectivity measures (*Lou et al., 2017*). When density and connectivity based preservation statistics are important

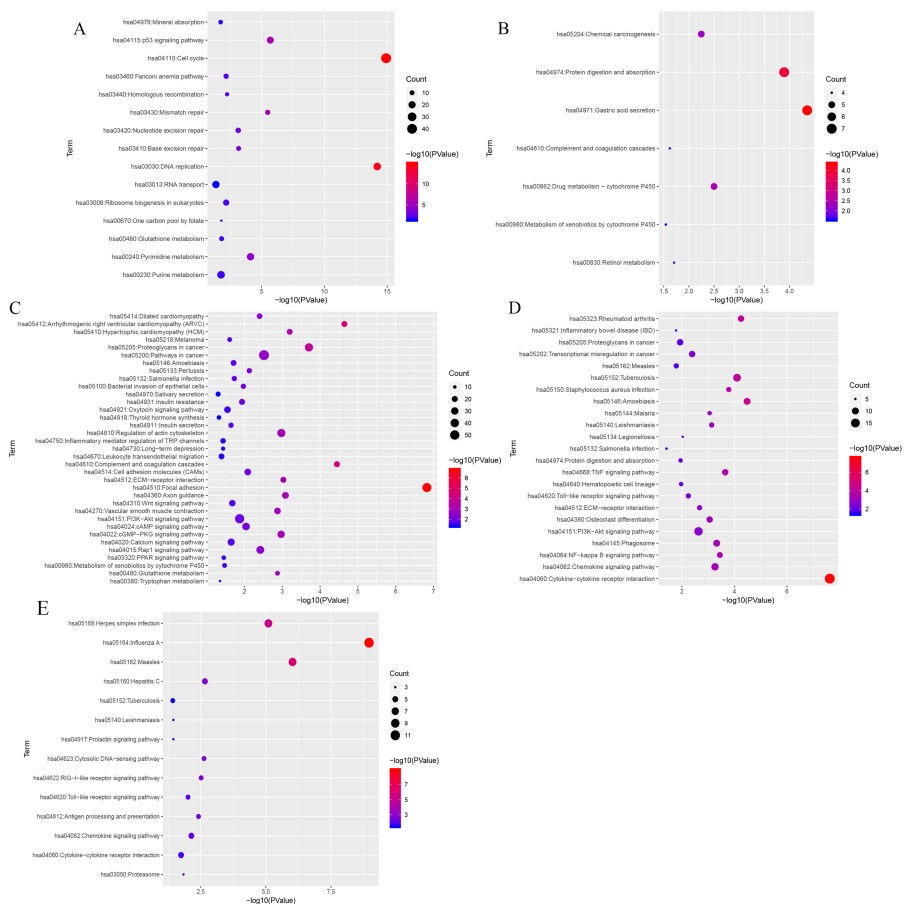

**Figure 7  KEGG pathway enrichment analyses of the (A) blue, (B) black, (C) turquoise, (D) greenyellow and (E) salmon modules.** The negative logarithm of the *p*-value is assigned to the *x*-axis and the term of each pathway to the *y*-axis. The size of the bubble shows the numbers of the genes enriched in each pathway, while the colors indicate the enrichment significance (from blue to red designated as less to high significance).

factors for judging the preservation of a network, Z summary score was preferentially selected to evaluate the preservation (*Langfelder et al., 2011b*). Although turquoise, blue and greenyellow had low preservation according to the median rank results, they are still considered to be conserved since the Z summary scores of these modules were 39, 52 and 22, respectively. Black module has relatively high Z summary score and low median rank, indicating high preservation. In the current study, all the four gene modules were considered to be conserved and selected for further analysis.

INHBA, Inhibin-$\beta$A (INHBA), a ligand belonging to the transforming growth factor-$\beta$ superfamily (*Oshima et al., 2014*), is associated with cell proliferation in various tumor types including colon adenocarcinoma (*Lin et al., 2020a*; *Miao et al., 2020*; *Miyamoto et al., 2020*), pancreatic cancer (*Liu et al., 2020*), gastric cancer (*Chen et al., 2019*) as well as oral squamous cell carcinoma (*Lin et al., 2020b*). Many studies have demonstrated the prognostic role of INHBA in colon adenocarcinoma (*Chen et al., 2020*; *Li et al., 2020*;
**Table 1  LASSO regression results.** Genes selected by the LASSO logistic regression, with the estimated coefficients and odds ratio.

| Gene | Coefficient | Odds ratio |
| --- | --- | --- |
| ADIPOQ | −0.16241431554 | 0.8500889265 |
| ARHGAP39 | 1.14238882470 | 3.134246595 |
| ATAD3A | 0.85913784838 | 2.361124169 |
| C1orf95 | −1.99447221881 | 0.1360854586 |
| CWH43 | −0.05325279770 | 0.9481402945 |
| GRIK3 | −4.02900881511 | 0.01779195633 |
| INHBA | 0.19110414846 | 1.210585526 |
| LYVE1 | −0.01065353474 | 0.9894030132 |
| RDH12 | −0.09270793497 | 0.9114596669 |
| SCNN1G | −0.02709362847 | 0.9732701115 |
| SIGLEC11 | −0.17337880264 | 0.84081905 |

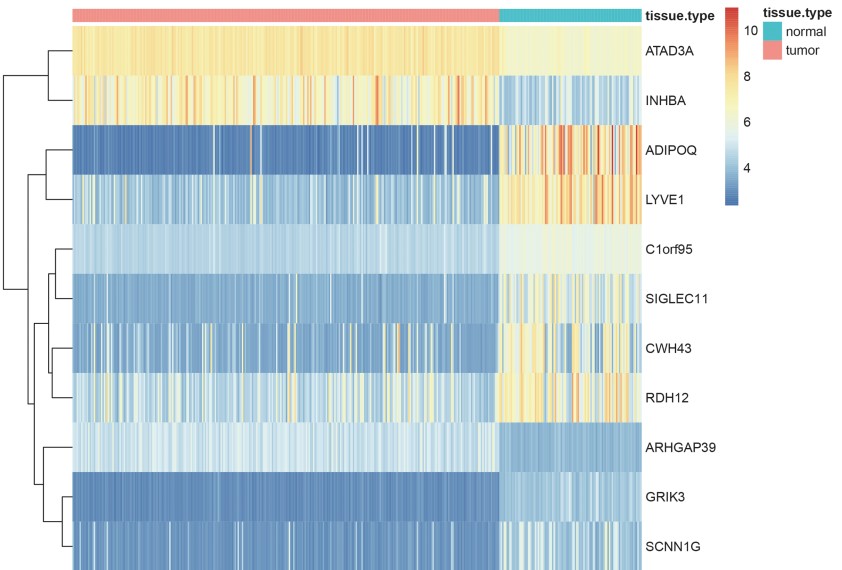

**Figure 8  Heatmaps showing the expressopms pf 11 hub genes between the gastric cancer patients and normal controls in training set.** The $x$- and $y$-axes present the samples and genes, respectively. In the $x$-axis, pink and green represent the gastric cancer and normal samples, respectively. The scale bar from blue to red represented low to high expressions of each gene in each sample.

*Miao et al., 2020*; *Sun et al., 2020*), and the role of INHBA in gastric cancer has been widely reported also. INHBA was highly expressed (*Kaneda et al., 2011*; *Seeruttun et al., 2019*; *Zhang et al., 2010*) and aberrantly methylated (*Zhang et al., 2019*) in gastric tumor samples, and high INHBA expression was associated with significantly poorer 5-year survival than low expression group (*Katayama et al., 2017*; *Wang et al., 2012*). One study has demonstrated that INHBA gene silencing could inhibit gastric cancer cell migration and invasion by impeding TGF-$\beta$ signaling pathway (*Chen et al., 2019*).

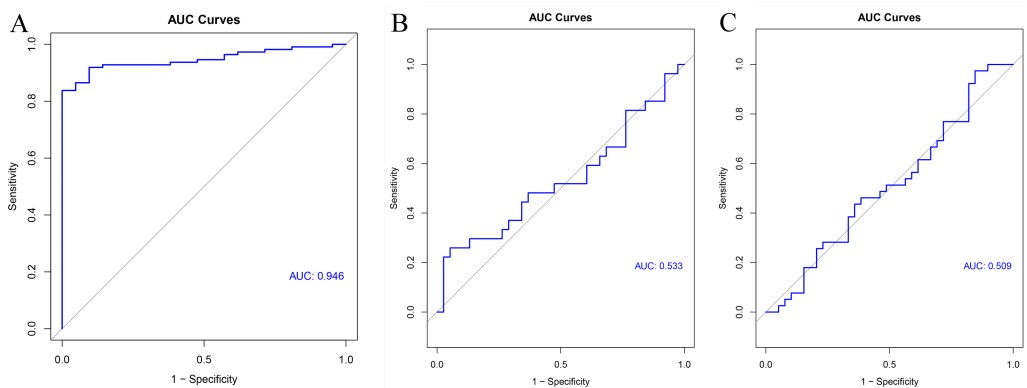

**Figure 9** **Validation results.** ROC curve of the classifier predicted by these 11 hub genes in the testing set of (A) gastric cancer, (B) colorectal cancer and (C) pancreatic cancer.

ADIPOQ is one of the most important adipocytokines secreted by adipocytes (*Parida, Siddharth & Sharma, 2019*), and the polymorphisms of *ADIPOQ* have been reported to correlate with several types of cancer, including colorectal (*Nimptsch et al., 2017*; *Tan et al., 2017*) and breast (*Mendez-Hernandez et al., 2017*) cancers. A study focusing on the molecular mechanisms ADIPOQ participated in has revealed that ADIPOQ induces cytotoxic autophagy in breast cancer cells through STK11/LKB1-mediated activation of the AMPK-ULK1 axis (*Chung et al., 2017*). Another study has reported that miR-370 inhibits the proliferation, invasion, and epithelial-mesenchymal transition of gastric cells by directly downregulating receptor 4 of ADIPOQ (*Feng et al., 2018*). Overexpression of another microRNA, miR-15b-5p, promotes the metastasis of gastric cancer by regulating ADIPOQ receptor 3 (*Zhao et al., 2017*).

ATAD3A is a nuclear-encoded mitochondrial enzyme, involving in mitochondrial dynamics, cell death, and cholesterol metabolism (*Teng, Lang & Shay, 2019*). It has been reported to correlate with hepatocellular carcinoma (*Liu et al., 2019a*) and breast cancer (*Daniel et al., 2019*), and it might be an effective therapeutic target in cancer treatment (*Teng et al., 2016*). *ATAD3A* is differentially expressed between paclitaxel-resistant and -sensitive MCF7 breast cancer cells (*Daniel et al., 2019*). A study has revealed that ATAD3A is upregulated in hepatocellular carcinoma and ATAD3A upregulation is correlated with poor prognosis (*Liu et al., 2019a*).

LYVE1 acts as a receptor and binds to both soluble and immobilized hyaluronan (*Banerji et al., 2016*), may function in lymphatic hyaluronan transport and tumor metastasis (*Wu et al., 2019*). The dysregulation of LYVE1 closely correlated with many types of tumor, like gastric cancer (*Ozmen et al., 2011*), colorectal cancer (*Gao et al., 2006*), breast cancer (*Kato et al., 2005*), lung cancer (*Koukourakis et al., 2005*) and liver cancer (*Mouta Carreira et al., 2001*). LYVE1 has been studied extensively for its possible role in cancer diagnosis and prognosis in cancer. One study has demonstrated that LYVE1 was upregulated in gastric cancer, and overexpression of LYVE1 positively correlated with perineural invasion and lymph node in gastric cancers (*Ozmen et al., 2011*). And the expression of LYVE1 might

be a biomarker to predict the existence of regional lymph node metastasis in early gastric cancer (*Fujimoto et al., 2007*).

RDH12, an NADPH-dependent retinal reductase, catalyzes the reduction of all-trans retinal to all-trans retinol (*Belyaeva et al., 2005*). It was significantly decreased in gastric tumor samples (*Kropotova et al., 2013*) and cervical squamous cell carcinoma samples (*Peng et al., 2015*). RDH12 was also one of the differentially expressed metabolism-related genes, and correlated with the prognosis of gastric cancer patients (*Wen et al., 2020*).

GRIK3 mainly participates in the neuroactive ligand–receptor interaction pathway, and GRIK3 upregulation is associated with poor survival in gastric cancer (*Gong et al., 2017*). GRIK3 promotes epithelial-mesenchymal transition by regulating the SPDEF/CDH1 signaling in breast cancer cells (*Xiao et al., 2019*).

There have been few studies focusing on the relationship between gastric cancer and *SCNN1G*, *ARHGAP39*, *C1orf95*, *CWH43* or *SIGLEC11*. *SCNN1G* is one of the genes significantly upregulated in Ewing's sarcoma and fibromatosis samples (*Sarver et al., 2015*). *ARHGAP39* mutations or variations in copy number or expression level were found in several types of tumor-like tissues from the central nervous system, skin, prostate, and gastrointestinal tract (*Nowak, 2018*). ARHGAP39 interacts with p53 and BAX, and decreased expression of ARHGAP39 increases cell proliferation, leading to tumorigenesis (*Jones, 2017*). C1orf95 is one of the uncharacterized proteins correlated with diverse human cancers (*Delgado et al., 2014*). Another study focusing on scleroderma patients demonstrated the involvement of C1orf95 in cancer incidence (*Xu et al., 2016*). Sialic acid-binding immunoglobulin-like lectin-11 (*SIGLEC11*) is a primate-lineage–specific receptor of human tissue macrophages, and it is also expressed in brain microglia (*Angata et al., 2002*; *Shahraz et al., 2015*). A missense mutation of *SIGLEC11* has been detected in pancreatic cancer patients (*Jones et al., 2008*), and SIGLEC11 was significantly upregulated in the poor prognostic group of pancreatic cancer patients (*Stratford et al., 2010*). CWH43 was correlated with tumorigenesis in thyrotropin-secreting pituitary adenomas (*Sapkota et al., 2017*). One meta-analysis also showed that CWH43 was differentially expressed between colorectal cancer and normal (*Chu et al., 2014*) samples.

All these results from the previous studies demonstrate that the hub genes identified in our study are closely correlated with gastric cancer and play important roles in cancer development, progression, or proliferation.

The significant module and hub genes identified in this study are biologically rational. First, the clinically significant module identified in our study bears strong preservation, implying that this clinically significant module is conservative and could also be reproduced in other datasets. Further, it suggests that that modules constructed by WGCNA are reliable. Second, most of the genes in the significant module were enriched for specific GO terms and KEGG pathways closely relating to stomach or cancer physiology. For instance, GO analysis demonstrated that most of the genes in the clinically significant modules were closely related to digestion, carbohydrate metabolic process, and gastric acid secretion, as well as cell division and cell cycle. KEGG enrichment analysis also indicated that most of the genes in the clinically significant module were implicated in gastric acid secretion, protein digestion and absorption, as well as glycerolipid metabolism and the p53 signaling pathway.

Third, all the hub genes identified in our study had previously been reported to relate to cancer. Moreover, several hub genes are implicated in metabolic processes, influencing the development and progression of gastric cancer. A previous study has demonstrated the association between metabolic syndrome and gastric cancer (*Li et al., 2018b*). A study has detected increased fatty acid oxidation in gastric cancer (*Lee et al., 2019*), and adipocytes fuel gastric cancer by mediating fatty acid metabolism (*Tan et al., 2018*). It may thus be inferred that these genes are genuinely the hub genes in charge of the key processes in gastric cancer, and they deserve a deeper analysis and validation. Finally, by using machine learning methods, the hub genes were demonstrated to effectively discriminate the gastric tumor samples from normal samples. In our study, the predictive effects of ANN method was evaluated by AUC values (*Huang & Ling, 2005*). Herein, the AUC value was >0.8, indicating the excellent predictive results. Furthermore, these 11-gene model might be the specific predictors for gastric cancer, since the AUC values of this predictive model were less than 0.8 in other tumor types including, colorectal cancer and pancreatic cancer. All the results indicated that the expression profiles of these 11 hub genes have excellent predictive effects when discriminating gastric cancer samples from normal samples.

However, our study has limitations. First, all the hub genes were identified and validated only through bioinformatics, and further exploration of the biological functions and molecular mechanisms of these hub genes both in vitro and in vivo is required. Second, due to the limited availability of the data, we did not differentiate between intestinal-type and diffuse-type gastric cancers. More data are needed to analyze and identify the hub genes between these two types of gastric cancer and normal samples.

In summary, through WGCNA, we identified 11 hub genes, which might serve as potential diagnostic and/or therapeutic biomarkers for gastric cancer. Profile data mining by bioinformatics analysis is an available method to find potential diagnostic or therapeutic biomarkers systematically. Nevertheless, further investigations about the molecular mechanisms in which these hub genes are involved are still needed to verify the involvement of these genes in gastric cancer. Our findings provide a better understanding of the molecular mechanisms and putative diagnostic or therapeutic biomarkers for gastric cancer.

### Funding
This work was supported by the Science & Technology Department of Sichuan Province funding project (No. 2016FZ0108, 2017FZ0104), the 1.3.5 project for disciplines of excellence, West China Hospital, Sichuan University (ZYJC18010). The funders had no role in study design, data collection and analysis, decision to publish, or preparation of the manuscript.

### Grant Disclosures
The following grant information was disclosed by the authors:
Science & Technology Department of Sichuan Province: 2016FZ0108, 2017FZ0104.

1.3.5 project for disciplines of excellence, West China Hospital, Sichuan University: ZYJC18010.

## Competing Interests

The authors declare there are no competing interests.

## Author Contributions

- Chunyang Li conceived and designed the experiments, performed the experiments, analyzed the data, prepared figures and/or tables, authored or reviewed drafts of the paper, and approved the final draft.
- Haopeng Yu performed the experiments, analyzed the data, prepared figures and/or tables, and approved the final draft.
- Yajing Sun performed the experiments, analyzed the data, prepared figures and/or tables, performed the double check of the results, and approved the final draft.
- Xiaoxi Zeng conceived and designed the experiments, performed the experiments, analyzed the data, authored or reviewed drafts of the paper, and approved the final draft.
- Wei Zhang conceived and designed the experiments, authored or reviewed drafts of the paper, and approved the final draft.

## Data Availability

R code and training set, pre-derivation analysis, and testing set raw data are available in the Supplemental Files.

## Supplemental Information

Supplemental information for this article can be found online at http://dx.doi.org/10.7717/peerj.10682#supplemental-information.

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
