# Peer review of "Identification of the hub genes in gastric cancer through weighted gene co-expression network analysis"

_PeerJ, doi:10.7717/peerj.10682_

## Round 0.1 · original submission · Major Revisions

The wording “validation” should be removed from the title.

Reviewer 1 ·

Basic reporting

The use of English language is adequate
Well referenced
Article is well structured, all the used data is available with online or in supplementary material
Self contained

Experimental design

It is an elegant computational approach to a biological question, within the journal scope
Correctly defined research question and knowledge gap
General methods used are standard, and those are well described

Validity of the findings

I am missing further results to completely agree on validity of the findings.

Additional comments

Major:
data collection and preprocessing: 3 datasets with the larger sample sizes have been selected to “avoid batch effects”. However there exist computational approaches to overcome batch effects when working with datasets from different sources (Removing Batch Effects in Analysis of Expression Microarray Data: An Evaluation of Six Batch Adjustment Methods, Plos One 2011). Using such methods, you could have chosen many datasets for the training set and that would have greatly enforced this research.

Specify whether the 3 selected datasets were normalized independently or not.

Module preservation: you need to further interpret figures and results from this section. It looks like bigger modules are more conserved, how do you explain that? In addition, how do you explain that modules preserved in GSE13911 are smaller that the same one’s identified in the training set? As those datasets were generated within the same platforms, detected genes should be the same.

Identification of clinically significant modules: I am worried about the methodology used here. You do obtain a Pearson’s correlation between each ME and clinical traits. However pearson correlation is used to analize the relationship between two quantitative variables, but clinical traits (cancer/healthy) variable looks categorical. Please clarify that, perhaps providing additional scatter plots or so.

Validation of the hub genes: figure 9 shows a great ROC curve using an independent gastric cancer dataset. However as you are addressing a gastric cancer specific hub genes signature, I am missing here a ROC curve obtained with a distinct type of cancer. In this way you could show that those hub genes are of great value for gastric cancer specifically. In the same sense, it might be interesting to show in fig 9 ROC curves obtained with another regression algorithm. Therefore, please add ROC curves to fig.9 to allow visual comparisons and strengthen your conclusions.

Minor:
L53 (gender ratio and age at diagnosis) : provide further details for readers

It would be great to further develop in introduction the use of regressions for such biological questions. In addition you don't specify in the intro that you are validating your model in an independent dataset at the end.

Justify the reasons why a simple glm regression algorithm was chosen instead of another available approach (neural nets, random forest etc) to validate you hub genes.

Reviewer 2 ·

Basic reporting

This paper describes a systematic strategy to discover putative biomarkers genes that can be used for the diagnosis of gastric cancer. This is an original paper focused on an important real problem. In addition, the manuscript is well written. So, it deserves to be published. However, several concerns must be corrected upon before Acceptance.

Title
I suggest eliminate the ¨validation¨ term from the title, because no experimental evidence is provided. Instead, the validation section corresponds to a statistical analysis used to test the reliability of the results, but it cannot be considered as a validation from the clinical or biological point of view.

Introduction
Line 141 change ¨types¨ by condition or a more appropriate term.

Material and methods
In this section must be defined the contrasting condition (e.g. type, stage).

Discussion
Lines 253-265 could be placed in the introduction. Instead, a brief paragraph to remember the subject of this study can be provided.

Experimental design

Results
- The Supplementary Figure 1 is not self-explained. This Figure must indicate the meaning of the color bar below the dendogram. Please, add a figure description.

- In the introduction, the authors mentioned that ¨There are two types of gastric cancer, diffuse and intestinal types…¨. However, they don´t indicate what kind of cancer type are the samples derived from. All the dataset employed come from the same cancer type?. Please specify.

- Please clarify why you don´t remove any sample from your dataset. What was your decision rule?.

- In Figure 2. Please enlarge the A and B images to clearly shown the plotted values. In Figure 2D, the R and slope values, must be plotted in the Figure, not in the title.


- In Figure 4A, please explicitly indicate the meaning of the scale in the heatmap (p-value). In Figure 4B, please explicitly provide the meaning of y axis (z-score), and indicate the meaning of error bar. What does mean the p-value of the title in Figure 4B?, please insert it in the image.

- Please provide a clear description for Supplementary Figure 2. Please, edit the image, indicating the cor and p value in the image, not in the title.

- In Figure 6, please enlarge the axis title of the image, because it is difficult to read. How the adjusted p value was estimated? I suggest just plot the meaningful enriched GO terms. Please provide a description for the threshold value.


- In the Supplementary table 2, I suggest remove all non-significative GO terms. In the description of Figure 7, please replace ¨in this¨ by in each. I suggest manually curate your annotation, since terms as ¨biosynthesis of antibiotics, not make me sense at all. Please edit the key to homogenize the format. In Figure A, count is above p-value and in B is upside down.

- Why did you plot the adjusted-p value for GO-enrichment analysis but the p-value for the KEGG analysis?

- In the Figure 9, I suggest to make a heatmap showing the expression profile of ¨hub genes¨, because the genes showed here are only a preliminary list of candidate genes that meet just one of the criteria taken into account and not correspond to the genes keep in the final list. Instead, this figure can be added as a Supplementary figure. If you decide maintain this figure, please indicate the meaning of the key scale and re-organize the clades in order to group the tissue condition to enhance the visibility of the clades in the x-axis. Another more simplistic option is provide a supplementary table with the FC of each gene and its associated FDR or adjusted p-value.

Validity of the findings

no comment

---

## Round 0.2 · Major Revisions

Please carefully revise according to the remaining reviewer's suggestions and comments.

Reviewer 1 ·

Basic reporting

OK

Experimental design

OK

Validity of the findings

OK

Additional comments

Identification of clinically significant modules: I still worried about the way you correlate a continuous variable and a categorical dichotomous variable. Because correlation talks about how much linear dependency is there between these two variables - if one variable increases whether another one increase or decrease- , your analysis is incorrect from a conceptual point of view. Furthermore, Spearson coefficient is used to correlate continous and a categorical which is ordinal/ranked, and so it is possible to obtain a linear relationship. Figure 5 A is definitely not a correlation matrix. And the pvalues obtained are ridiculously low, which is a bad sign. Please delete this correlation analysis and figure 5A from the manuscript.
In contrast, Figure 5B and its methodology are apropiate (since here you use a non-linear regression, which is oposite with your correlation analysis). Pvalues obtained in fig. 5B look more apropiate, and, in fact, they differ from figure 5A (black module pvalue is lower here than the blue module one, in contrast with figure 5A where it is not).
-
"l261: Using the screening criteria of |GS| > 0.75 and |MM |> 0.8, 148 genes were identified"
I dont understand your GS criteria; in fig 5B, maximum GS is close to 0.5

ROC curve figure will be much more informative if you integrate S fig3 curves, so visual comparison is straightforward. In addition, why didnt you perform cross-validation in the ANN regression to overcome overfitting here?

---

## Round 0.3 · accepted · Accept

The manuscript has been improved.

Reviewer 1 ·

Basic reporting

All the potential improvements have been fully adressed

Experimental design

All the potential improvements have been fully adressed

Validity of the findings

All the potential improvements have been fully adressed